# OpenReview forum: "Omni-SafetyBench: A Benchmark for Safety Evaluation of Audio-Visual Large Language Models"
_ICLR.cc/2026/Conference — ICLR 2026 Conference Withdrawn Submission_

### Official Review · Reviewer_UneT · 2025-10-30

**Soundness:** 2
**Presentation:** 3
**Contribution:** 2
**Rating:** 2
**Confidence:** 3

**Summary:**

The paper introduces Omni-SafetyBench, a new large-scale benchmark for evaluating the safety of Omni-modal Large Language Models (OLLMs). The benchmark is constructed by systematically transforming a seed set of 972 harmful text prompts into 24 different modality variations, including unimodal (image, video, audio), dual-modal (e.g., image-text), and omni-modal (e.g., image-audio-text) inputs. The authors propose two key metrics: a Safety-score, which is conditioned on the model's comprehension of the prompt to avoid inflated scores from misunderstanding, and a Cross-Modal Safety Consistency score (CMSC-score) to measure how robust a model's safety defenses are across different input modalities. The paper presents an extensive evaluation of 10 state-of-the-art OLLMs, revealing significant vulnerabilities, especially with complex audio-visual inputs. Furthermore, it analyzes the effectiveness of existing safety alignment techniques, concluding that they struggle with the out-of-distribution challenges posed by the diverse modality combinations in OLLMs.

**Strengths:**

- This is arguably the first benchmark to systematically evaluate OLLM safety across such a wide range of modalities (24 parallel variations), including joint audio-visual inputs. The scale of the benchmark (over 23,000 test instances) is commendable.

- The paper correctly identifies that a model's failure to understand a complex multimodal prompt can be mistaken for a successful safety refusal. The proposed Conditional metrics (C-ASR, C-RR) and the resulting Safety-score are a sensible and important methodological contribution for evaluating advanced multimodal models. The CMSC-score is also a novel and useful way to quantify the consistency of safety defenses.

- The authors provide a thorough evaluation of 10 different OLLMs, offering a valuable snapshot of the current landscape. The subsequent analysis of both inference-time and post-training alignment methods provides concrete evidence for the challenges in making OLLMs robustly safe, particularly the out-of-distribution generalization problem.

**Weaknesses:**

- The core weakness is the paper's limited conceptual novelty. The paradigm of creating a safety benchmark by curating or generating harmful prompts is well-established. This work extends this paradigm to more modalities. While a necessary and useful resource, it does not fundamentally change how we approach safety evaluation. The work contributes to the cat-and-mouse dynamic of benchmark creation and model patching, which has questionable long-term scientific value.

- The entire benchmark is derived from a single seed dataset via automated modality transformation (e.g., text-to-speech, text-to-image). This approach, while systematic, may result in attack patterns that are syntactically diverse but semantically shallow. The attacks primarily consist of placing explicit harmful text into different media formats. This may not capture more sophisticated, inherently multimodal safety risks where harm arises from the subtle interaction of seemingly benign elements across modalities, rather than the direct transcription of a harmful phrase.

- The central findings, i.e., closed-source models are generally safer than open-source ones, and that model safety degrades as input complexity increases, are consistent with prior literature and the general intuition of the field. While the paper provides strong quantitative evidence for this in the OLLM context, the insights themselves are not particularly surprising, which lessens the paper's overall impact.

**Questions:**

- Could the authors elaborate on the conceptual contribution of this work beyond providing the benchmark for omni-modal settings? What new scientific understanding about multimodal safety does this benchmark enable, which previous paradigms could not?

- The attack generation is fully automated from textual seeds. This seems to test a model's robustness to recognizing a core harmful concept across different encodings. Have you considered attacks where the harmful instruction is emergent from the combination of modalities, rather than being explicitly present in each?

- Given that the benchmark generation process is deterministic from a public dataset, it seems susceptible to being solved by training models specifically on these types of transformations (e.g., recognizing harmful text in typographic images). How do the authors view the long-term utility of Omni-SafetyBench in this context?

---

### Official Review · Reviewer_ANrB · 2025-10-30

**Soundness:** 3
**Presentation:** 3
**Contribution:** 2
**Rating:** 6
**Confidence:** 3

**Summary:**

This paper introduces Omni-SafetyBench, the first comprehensive parallel benchmark designed specifically for evaluating the safety of Omni-modal Large Language Models (OLLMs)—models that integrate text, visual, and auditory processing. Addressing the gap in existing benchmarks (which lack coverage of audio-visual joint inputs and cross-modal consistency evaluation), Omni-SafetyBench includes 23,328 test samples across 24 modality variations (encompassing unimodal, dual-modal, and omni-modal paradigms, with dedicated audio-visual harmful cases) derived from 972 seed entries from MM-SafetyBench. The authors propose two tailored metrics: Safety-score (calculated from Conditional Attack Success Rate (C-ASR) and Conditional Refusal Rate (C-RR), accounting for model comprehension of complex inputs) and Cross-Modal Safety Consistency (CMSC)-score (measuring safety consistency across modalities).

**Strengths:**

- **Potentially Useful Benchmark**: Omni-SafetyBench fills the gap in OLLM safety evaluation by being the first benchmark to focus on audio-visual joint inputs and cross-modal safety consistency. The modality variations (unimodal: text/image/video/audio; dual-modal: image-text/video-text/audio-text; omni-modal: image-audio-text/video-audio-text) provide a comprehensive testbed for OLLMs’ multi-modal safety.
- **Comprehension-Aware and Consistency-Focused Metrics**: The proposed Safety-score addresses the flaw in prior evaluations (inflated safety scores due to poor model comprehension of complex inputs) by first measuring comprehension via question-answer pairs and only evaluating safety on well-understood inputs. The CMSC-score, which quantifies cross-modal consistency using standard deviation of Safety-scores, is essential for identifying vulnerabilities where attackers could exploit modality switching to jailbreak OLLMs.
- **Extensive Experiments**: The authors evaluate 10 state-of-the-art OLLMs nd categorize them into four safety profiles (“Robustly Safe,” “Selectively Risky,” “Consistently Risky,” “Critically Unstable”), providing clear actionable insights. Additionally, testing existing safety alignment methods (ECSO, CoCA, ShiftDC for inference; HH-Harmless, PKU-SafeRLHF, VLGuard, SPA-VL for post-training) reveals challenges in OLLM safety alignment, guiding future research directions.

**Weaknesses:**

**Dependence on Qwen-Plus as Judge Model**: While the authors validate Qwen-Plus’s consistency with human annotators (overall accuracy >0.9), relying on a single closed-source judge model introduces potential bias, as different judge models (e.g., GPT-4o, Claude-3.5) may have varying standards for “comprehension,” “harmful content,” or “refusal,” which could affect Safety-score and CMSC-score calculations.

**Incomplete Analysis of Modality-Specific Vulnerabilities**: While the paper identifies that audio-visual inputs trigger the most vulnerabilities, it provides limited granularity on why specific modalities (e.g., typographic vs. diffusion images, TTS vs. TTS+noise audio) are more impactful. For example, the authors note that noise reduces audio defenses but do not explore the mechanism (e.g., noise disrupting audio-text alignment) or whether this generalizes to other noise types.

**OLLM Safety Methods**: It would be great if the authors could compare with methods like [1][2][3]

[1] Uniguard: Towards universal safety guardrails for jailbreak attacks on multimodal large language models

[2] Mllm-protector: Ensuring mllm's safety without hurting performance

[3] Jailguard: A universal detection framework for llm prompt-based attacks

**Questions:**

See weaknesses

---

### Official Review · Reviewer_1Bd4 · 2025-11-01

**Soundness:** 2
**Presentation:** 2
**Contribution:** 2
**Rating:** 2
**Confidence:** 3

**Summary:**

The paper introduces Omni-SafetyBench, the first benchmark for evaluating the safety of OLLMs that process text, image, video, and audio jointly. It uniquely assesses cross-modal safety consistency and audio-visual harm cases. Two new metrics are proposed: Safety-score and CMSC-score, which enable nuanced evaluation under complex multimodal inputs. The key findings: safety degrades sharply with multimodal complexity, and only three models exceed 0.6 in both metrics. Further analysis shows inference-time defenses are limited and post-training methods suffer from out-of-distribution issues. Overall, the work highlights the urgent need for robust safety alignment in OLLMs.

**Strengths:**

*The paper identifies the safety of OLLMs as a critical and underexplored research challenge, providing valuable insights that could advance future work in multimodal AI safety.

* The introduction and experimental results are presented in a clear, logical, and well-structured manner, effectively guiding readers through the motivation, methodology, and findings.

* The dataset construction demonstrates a cost-effective and scalable strategy by extending existing resources to create diverse multimodal and parallel safety test cases.

**Weaknesses:**

* The definitions and implementation details of C-ASR and C-RR are unclear, particularly regarding how the authors determine when a model “understands” an input or produces a "safe" response. This lack of transparency raises concerns about reproducibility and makes it difficult to interpret the reported safety scores with confidence.

* The motivation and theoretical justification for the CMSC-score are insufficient. It is not well explained why the standard deviation across subcategories serves as an appropriate measure of cross-modal consistency.

* The data construction process for generating new modality variants appears to provide redundant information for new modalities, such as audio. Consequently, the experiments provide limited insight into why increasing modality combinations with equivalent semantic content may lead to decreased OLLM performance.

**Questions:**

* In Section 4.2, how does the experimental data address OOD problems? What evidence is provided to confirm that the test data are indeed OOD?

* Why is the unique information contributed by new modalities not considered in this work, given that such information is common in real-world scenarios?

---

### Official Review · Reviewer_4UQ5 · 2025-11-12

**Soundness:** 1
**Presentation:** 2
**Contribution:** 1
**Rating:** 2
**Confidence:** 5

**Summary:**

This paper presents Omni-SafetyBench, the first safety-evaluation benchmark for audio–visual–text joint inputs to OLLMs, and introduces two new metrics: Safety-score and CMSC-score.

**Strengths:**

1. It is the first safety benchmark to cover audio-visual-text joint inputs, filling the modality-coverage gap highlighted in Table 1 and furnishing the community with a foundational dataset.

2. The paper introduces a comprehension-aware safety philosophy that assesses safety only after understanding, which is more realistic for human-computer interaction than simply reporting ASR, even if the implementation is flawed.

3. The experimental scale is solid: 23,328 samples across 10 models with multiple metrics; results are systematically presented (Tables 3–12) and full prompts are released (Appendix E.4), ensuring reproducibility.

**Weaknesses:**

1. The paper treats "not understood" as "excluded from safety evaluation," yet it relies solely on an LLM-as-judge to decide whether an input is understood. Appendix E.2 shows this judge misclassifies understanding at a rate of 11 percent, and the authors provide no robustness check such as a sufficient human validation sample, so C-ASR and C-RR are systematically biased.

2. Every audio-visual sample is generated automatically from seed text taken from MM-SafetyBench. The authors never verify cross-modal semantic fidelity, for example whether the diffusion image faithfully conveys the meaning of "homemade weapon." Appendix B contains no alignment-quality metric, so an "attack failure" may simply reflect modal distortion rather than model defense.

3. Omni-modal inputs perform worse than unimodal ones, but the paper offers no way to tell whether this is caused by "more modalities" per se or by redundancy, conflict, or reduced information density. For instance, a typographic image plus TTS audio repeat the exact same content and may distract attention instead of testing joint reasoning safety.

4. Safety-score rewards refusal (C-RR), yet the authors never measure the false-refusal rate on benign multimodal inputs, for example a normal video with an innocuous voice-over. A model can easily game the metric by adopting an overly conservative policy that sacrifices usability.

5. Post-training uses only text or text-image data, but the benchmark demands generalization to audio-visual-text. The experiment therefore tests out-of-distribution robustness rather than the alignment method itself. If the claim is "cross-modal alignment," the paper should propose a cross-modal training recipe such as a modality-transfer alignment loss; otherwise the attribution is invalid.

6. The binary safe/unsafe label is too coarse. Outputs that first list steps and then discourage execution are labeled unsafe (see Fig. 10 Dual-modal), even though harmful intent and harmful execution are not separated, ignoring the model's gradual safety mechanism.

**7. The paper claims to evaluate OLLMs, but the claim is inaccurate. Most models tested cannot accept all modalities as a single joint input. For example, VITA-1.5 can process text+image or audio+image, but not text+audio+image simultaneously, so the design of Table 3 is fundamentally flawed.**

**Questions:**

None.

---

### Note · Authors · 2025-11-12

I have read and agree with the venue's withdrawal policy on behalf of myself and my co-authors.